# Sodium Fluorescein-Guided Surgery for Resection of Brain Metastases from Lung Cancer: A Consecutive Case Series Study and Literature Review

**DOI:** 10.3390/cancers15030882

**Published:** 2023-01-31

**Authors:** Xing Cheng, Jie Chen, Ronghua Tang, Jian Ruan, Deqiang Mao, Haifeng Yang

**Affiliations:** Department of Neuro-Oncology, Chongqing University Cancer Hospital, Chongqing 400030, China

**Keywords:** sodium fluorescein, brain metastases from lung cancer, fluorescence-guided surgery, extent of resection

## Abstract

**Simple Summary:**

Surgical resection still plays an important role in the treatment of lung cancer with brain metastases. Accurately identifying the border between normal brain tissue and tumor invasion under the microscope to maximize the extent of resection without causing neurological dysfunction is critical but still challenging. Here, we introduced the utilization of sodium fluorescein in the surgical resection compared with the previous studies, focusing on brain metastases from lung cancer, the most common secondary malignant brain tumors, and included the control group. This study will provide a valuable reference for the precise surgical treatment of brain metastases from lung cancer.

**Abstract:**

(1) Introduction and objective: Surgical resection plays an important role in the multidisciplinary treatment of lung cancer patients with brain metastases (BMs). Precisely distinguishing the tumor border intraoperatively to improve and maximize the extent of resection (EOR) without causing permanent neurological defects is crucial but still challenging. Therefore, we introduced our experience of utilizing sodium fluorescein (SF) in microneurosurgery of BMs from lung cancer. This study aims to evaluate whether the use of SF-guided surgery has a positive impact on postoperative outcomes. (2) Materials and methods: A retrospective study was performed to collect data on a consecutive case series of patients with BMs from lung cancer who underwent surgical resection from January 2020 to December 2021 at the Department of Neuro-Oncology, Chongqing University Cancer Hospital. A total of 52 patients were enrolled, of which 23 received SF-guided surgery and 29 did not. EOR was assessed pre- and postoperatively on T1 contrast-enhanced MRI. Clinical and epidemiological data as well as follow-up were gathered and analyzed. (3) Results: Compared with the non-SF-guided group, the SF-guided group revealed a significantly better EOR (87.0% vs. 62.1%) and a lower incidence of local recurrence (8.7% vs. 34.5%). Survival benefits were seen in patients with NSCLC, patients who were undergoing SF-guided surgery, and patients receiving postoperative systemic therapy. (4) Conclusions: SF-guiding under the YELLOW 560 nm filter is a safe and feasible tool for improving the EOR in patients with BMs from lung cancer, leading to better local recurrence control and prolonged survival.

## 1. Introduction

Brain metastases (BMs) are one of the most common tumor types in the adult central nervous system, accounting for about 20% of central nervous system malignancies [1]. Among all malignancies, lung cancer, breast cancer, melanoma, and colorectal cancer are prone to BMs, while BMs from lung cancer are the most common type [1,2]. The brain is not only the prone site of metastasis at the initial diagnosis of lung cancer patients (10–20%), but also about 20–65% of patients are reported to have BMs during the disease [3,4]. BMs can cause various symptoms, including headache, vomiting, and hemiplegia, severely reducing the quality of life and shortening the survival period of patients [2,5].

Although the success of targeted therapy and immunotherapy has made significant changes in the treatment of lung cancer in the past decade, many BMs patients still fail to benefit due to various factors such as the blood-brain barrier (BBB), cystic lesions, and drug resistance [6]. Surgical resection, therefore, remains one of the crucial treatments for BMs patients. Overall, surgery has the advantages of rapid relief of increased intracranial pressure and reduction of tumor burden [1,3], confirming the pathological diagnosis [3], the anti-seizure effect [2], and reducing the use of corticosteroids [5,6], thus, leading to prolonged overall survival (OS) [1,2,3,5].

In general, the surgical-oncological objective is to improve and maximize the extent of resection (EOR) while reducing the risk of new neurological deficits postoperatively, which affects the quality of life (QoL) and survival of patients a lot [7,8]. Although various supportive approaches and technologies have been developed to accommodate the aim, such as intraoperative monitoring, stimulation, awakening, intraoperative ultrasound, navigation, and MRI [9,10], this concept retains technical restrictions as access to such intraoperative techniques is limited in some centers. Furthermore, despite BMs being reported to be characterized by a well-established dissection plane between the tumor and the surrounding brain [7], it can be difficult to distinguish healthy brain tissue from tumor-infiltrated tissue when surgery is performed under white light, which is the most common technique in many centers (Figure 1a).

Recently, there have been increasing reports of fluorescence-guided surgery (FGS) in neurosurgical practice, particularly in high-grade gliomas (HGGs), as a tool to visualize the pathological and non-pathological normal brain parenchyma intraoperatively, which is conducive to maximizing EOR [11,12]. However, the majority focused on 5-aminolevulinic acid (5-ALA) as its only approved drug for fluorescence-guided glioma surgery by FDA [13]. Despite surgical benefits, 5-ALA also has some disadvantages, such as high price ($200 per patient in China), only oral administration 3–5 h before surgery (non-repeatability), the following phototoxicity within 24 h after administration, and inconvenience of requiring a total darkening of the operating room, which limits its clinical utility [11,12,13]. In addition, unlike in HGGs, 5-ALA was reported not to have an essential role in the resection of BMs, at least not recommended for routine, due to its low rate of fluorescence enhancement or even poorer patient survival [14,15]. These limitations have prompted neurosurgeons to seek cheaper, more convenient, and less toxic alternatives to 5-ALA, especially for BMs.

Although not yet approved by the FDA, increasing applications of sodium fluorescein (SF/Na-Fl) have been reported currently in multiple brain tumor surgeries (even off-label use), which is equally effective in tumor staining, but inexpensive (approximately 1/15 the price of 5-ALA), more convenient, and has fewer side effects when compared to 5-ALA [16,17,18,19]. However, to our knowledge, few clinical studies have introduced and discussed the use of SF in the neurosurgical removal of BMs [18,20,21,22,23,24,25], among which only three studies compared SF-guided directly to white light surgery [18,23,24], and no study was particularly focused on lung cancer and included a control population.

Hence, we retrospectively analyzed our experiences with SF in patients undergoing surgical resection of BMs from lung cancer. This study aimed to investigate the utility of SF in improving the EOR of BMs from lung cancer and to assess its impact on postoperative sequelae and overall outcome.

## 2. Materials and Methods

### 2.1. Patients

After obtaining approval from the Medical Ethics Committee of Chongqing University Cancer Hospital (CUCH), a retrospective cohort review was performed to collect data on a consecutive case series of adult (age greater than or equal to 18 years) patients with BMs from lung cancer who underwent surgical resection from January 2020 to December 2021 at the Department of Neuro-Oncology, CUCH (Appendix A). Patients were dichotomized according to whether the SF was used or not. Sodium fluorescein was introduced in the department in November 2019 and gradually became an optional tool, but not a standard approach for neurosurgeons either used in HGGs or BMs surgical resection after being approved by the Pharmacy Administration Committee and Medical Ethics Committee of CUCH. Whether to conduct SF-guided surgery in BMs patients mainly depended on the willingness and informed consent of patients and the judgment of neurosurgeons. Informed consent was obtained from all patients for the off-label use of SF. All the fluorescent and non-fluorescent surgeries were performed by the same three neurosurgeons.

Inclusion criteria: all first-time BMs patients identified within the time period obtained surgical-histological evidence of metastasis from lung cancer.

Exclusion criteria: cases with recurrent BMs, cases with BMs from other malignancies, and cases where postoperative MRI results were not available.

### 2.2. Pre- and Postoperative Clinical and Radiological Assessment

All patients were evaluated with gadolinium-enhanced magnetic resonance imaging (MRI) of the brain to identify tumor location and number of metastases (Figure 1e,f), and systemic PET/CT to assess extracranial metastasis before the operation. Contrast-enhanced MRI was performed within 72 h postoperatively to evaluate hemorrhage and to confirm the EOR (Figure 1g,h). Demographic and clinical characteristics of all patients who met the study criteria were recorded, including age, sex, preoperative clinical performance scores (Karnofsky Performance Status, KPS; 0-100), and postoperative therapy.

### 2.3. Surgical Protocol

SF was injected intravenously, at a dose of 5 mg/kg, by a central venous line in the operatory room, immediately upon completion of the induction of general anesthesia. A skin test was administered 24 h before to confirm a negative result of allergies. All surgical interventions were performed using a dedicated surgical microscope (OPMI PENTERO 900 with a YELLOW 560 nm filter, Carl Zeiss Meditec^®^, Jena, Germany). Neuronavigation was used only for craniotomy planning, and cerebrocortical tumor localization, but not for resection control in fluorescence-guided patients. The surgical aim was gross total resection (GTR) in both fluorescence-guided and non-fluorescence-guided patients if realizable. Surgical resection was performed en bloc when feasible, otherwise piecemeal until all fluorescent tissue was removed. As the surgery progressed, the neurosurgeon could switch to different light modes depending on the surgical requirements. In cases with more than one lesion, the goal was resection of the largest and/or symptom-giving one. Other technical adjuncts (intraoperative ultrasound, neurophysiological monitoring, etc.) were available when needed.

### 2.4. Endpoints

The primary endpoint was the proportion of patients with a GTR of the contrast-enhanced lesion. EOR was evaluated based on the postoperative MRI using the Response Assessment in Neuro-Oncology Brain Metastases (RANO-BM) criteria and sorted into three groups: measurable residual (>10 mm contrast-enhancing tumor remnant), unmeasurable residual (≤10 mm contrast-enhancing tumor remnant), or no residual (no contrast enhancement) [18,26]. The secondary endpoints were local recurrence and OS (censored 31 December 2022).

### 2.5. Statistical Analysis

Categorical variables were summarized as frequencies or proportions and were analyzed using chi-squared tests between the SF-guided group and the non-SF-guided group when applicable. P-values with corresponding chi-square values (χ^2^) were presented and *p* < 0.05 was considered to be statistically significant. Data processing and statistical analysis were performed by using IBM SPSS Statistics software (version 27.0, IBM Corp., Armonk, New York, NY, USA).

The Kaplan–Meier method and the log-rank test were used for the univariant analysis of the OS, concerning sex, age, KPS, tumor location, number of intracranial metastases, extracranial metastasis or not, SF-guided or not, EOR, pathological type, postoperative systemic treatment, postoperative brain radiotherapy, and postoperative local recurrence. Variables that were observed with significant differences between two subgroups in the univariate analysis were selected for multivariate analysis. The Cox proportional hazards model was used to determine associated hazard ratios (HR) and their 95% confidence intervals. Here, OS refers to the period from the operation to the patient’s death or last follow-up.

## 3. Results

### 3.1. Descriptive Parameters

Fifty-two patients (38 male, 14 female) ranging from 26 to 77 years old, with a median age of 56 years, were enrolled. SF-guided surgery was performed in 23 cases (44.2%). Median preoperative KPS amounted to 80 (ranging from 40 to 90). Both the SF-guided and non-SF-guided groups were comparable in terms of sex, age, preoperative KPS, tumor location (supratentorial or infratentorial), number of intracranial metastases, extracranial metastasis or not, pathological type (NSCLC or SCLC), postoperative systemic treatment (chemo-/targeted/immunotherapy), and postoperative brain radiotherapy; no statistically significant difference was found (Table 1). 

### 3.2. Fluorescence of BMs

In the SF-guided group, after approximately a one-hour course of craniotomy, all tumors had a relatively homogeneous and moderate yellow fluorescence under the yellow 560 nm filter (Figure 1b). Fluorescence was sensitive to visualize the tumor border (white dashed frame), and even distinguish the draining veins (blue arrowheads), which was conducive to protecting the cerebral cortex. Tumor fluorescence maintained stability for almost 3–4 h on average and no adverse effects, anaphylactic reactions, or postoperative neurological deterioration related to the SF-guided surgery were observed.

### 3.3. Surgical Resection and Local Outcome

In the SF-guided group, the early postoperative MRI demonstrated that 20 out of 23 patients (87.0%) had no residual tumor (Figure 1c,d), and only 3 (13.0%) had unmeasurable or measurable tumors. However, in the non-SF-guided group, 18 out of 29 patients (62.1%) had no residual tumor and 11 (37.9%) had unmeasurable or measurable tumors, revealing a significantly better EOR rate of the SF-guided group than that of the non-SF-guided group (Table 2). Similarly, although there was no significant difference between the two groups concerning the postoperative systematic treatment and brain radiotherapy (Table 1), only 2 patients in the SF-guided group had local recurrence, while 10 patients in the non-SF-guided group had local recurrence, which was significantly higher than that in the SF-guided group (Table 2).

### 3.4. Survival Analysis

Although the univariant analysis showed that sex, age, pathological type, SF-guided surgery, postoperative systemic treatment, and postoperative local recurrence were risk factors for patients’ survival, the multivariate analysis only indicated the pathological type, postoperative systemic treatment, and SF-guided surgery had an impact on the OS (Table 3).

Survival benefits were seen in patients with NSCLC, patients who were undergoing SF-guided surgery, and patients receiving postoperative systemic therapy (Figure 2), whereas sex, age, preoperative KPS, tumor location, number of intracranial metastases, extracranial metastasis or not, EOR, postoperative brain radiotherapy or not, and postoperative local recurrence did not influence survival significantly.

## 4. Discussion

Our retrospective study demonstrates that SF-guided resection of BMs from lung cancer is feasible, safe, and allows for a high rate of complete resection at early postoperative MRI, and no specific adverse event related to fluorescein was registered.

### 4.1. The Role of Surgery in the Treatment of Lung Cancer with BMs

Recently, with the increasing incidence and the development of diagnosis and treatment of lung cancer, the survival time of patients has been significantly prolonged, which also leads to an increased prevalence of brain metastases from lung cancer by year [1,2,27]. Although recent developments in targeted therapy and immunotherapy and stereotactic radiosurgery have provided a variety of options for patients with BMs from lung cancer, surgical resection plays an important role in properly selected patients, for its achieving the sample for diagnosis, alleviating the mass effect, and providing an opportunity for follow-up treatments [7,22]. Based on our knowledge and experience, standardly accepted indications for craniotomy include but are not limited: (1) bulky metastases (typically >3–4 cm in maximal unidimensional size) with suitable location and easy to resect, (2) life-threatening cerebral herniation caused by severe tumor edema or stroke, (3) suspected BMs from lung cancer but with pathological uncertainty, and (4) cystic lesions assessed may be insensitive to chemoradiotherapy or targeted therapy [2,27].

### 4.2. Fluorescence-Guided Agents for BMs Surgery

Since BMs and their recurrence can remarkably decrease the patients’ QoL and prognosis [5,18,24], radical resection is strongly encouraged, but not at the expense of causing permanent neurological defects, making dissecting the precise border of the tumor a crucial but challenging task. Tumor fluorescence visualization is currently an optional tool for neurosurgeons to aim the requirements, among which 5-ALA, SF, and indocyanine green (ICG) are the most commonly used fluorescent agents in neurosurgery [11,16,28]. Regarding 5-ALA, due to its low rate of fluorescence enhancement (only 66% of the tumors exhibited visible fluorescence) [14], is not reliable to use in BMs and therefore is not routinely used, while ICG is mainly used in surgery of vascular neoplasms such as hemangioblastoma [28].

Contrary to 5-ALA, SF is an unspecific fluorescent dye of a disrupted BBB. The mechanism of action for SF is different from 5-ALA as it is not accumulated intracellularly in tumor cells, but rather is distributed and accumulated in the extracellular space throughout brain areas where the BBB is disrupted, e.g., by BMs infiltration [29]. The intact BBB will prevent the overflow of SF; however, the presence of tumors will damage the BBB, which enables the obtaining of a reliable visualization of the tumor. SF was first reported to be used for fluorescence imaging of several brain tumors in 1948 by Moore et al. [30]. Since then, the literature on the application of SF in neurosurgical tumor resection has gradually increased, including HGGs, BMs, lymphoma, pituitary tumor, meningioma, medulloblastoma, etc. [16,17,18,19,31,32]. As for BMs, SF-guided resection has been reported to provide a better distinguishing tumor visualization, hence increasing EOR, reducing recurrence, improving postoperative QoL, and prolonging patient survival, when compared to conventional surgery [18,20,21,22,23,24,25]. In this retrospective study, we compared the impact of SF on resection extent, local recurrence control, and survival in two groups of patients with BMs from lung cancer and revealed similar results indicating that SF-guiding made the surgery easier and results in a higher EOR, lower local recurrent incidence, and better survival compared to the patients who received non-SF-guided surgery.

### 4.3. Prognostic Factors Affecting the Survival of Patients with BMs

Many factors could affect the prognosis and survival of lung cancer patients with BMs, the present study showed that SCLC was the most significant risk factor. Although the overall incidence rate of SCLC was low (13.46%, 7/52 in our cohort), due to its high malignancy and early metastasis, the prognosis was not as good as that of NSCLC despite intensive treatment [33]. The postoperative 1-year and 2-year cumulative survival rates of SCLC patients in our study were 42.86% (3/7) and 0% (0/7), respectively, which were far lower than NSCLC patients (85.11% (40/47) for 1-year and 25.53% (12/47) for 2-year survival). Another important factor affecting the prognosis of patients with BMs from lung cancer shown in the present study was postoperative systemic treatment. The average survival time for those who received the postoperative systemic treatment was 21 months, compared with 17 months for those who did not. Given that systemic or multiple intracranial metastases often occured in lung cancer patients with BMs, neurosurgical resection only alleviated the local symptoms of patients in the short term, gaining opportunities for follow-up treatment, but not determining the long-term prognosis; therefore, postoperative comprehensive treatment determined the OS of patients to a large extent [2,4]. Based on our present data and experience, we strongly recommend that all patients with BMs from lung cancer receive proper systemic therapy early after surgery to control the disease and prolong survival.

Interestingly, our study demonstrated sex, age (particularly when categorizing the patients into groups younger or older than 60 years), preoperative KPS (≥70 or <70), tumor location, number of intracranial metastases, extracranial metastasis or not, EOR, postoperative brain radiotherapy or not, and even postoperative local recurrence were not associated with patient survival. Different literature may have different conclusions on whether the variables above affect the prognosis of patients based on their research limitations [18,20,22,23,25], demanding well-designed prospective research for further assessment.

Finally, as we mentioned above, the SF-guided group was found to have a statistically significant improved OS both in the univariant and the multivariate analysis compared to the non-SF-guided group in the present study. This might properly be the result of a combination of multiple factors, such as the number of intracranial metastases, extracranial metastasis, EOR, and local recurrence. Since this was retrospective case data, the role of sodium fluorescein in improving the survival of patients still needs further study and demonstration. However, benefiting from the fluorescence, neurosurgeons could better distinguish the border of the tumor, not only minimizing tumor residual but also protecting the normal brain tissue, avoiding causing neurological defects, and improving the postoperative QoL of patients.

## 5. Conclusions

This retrospective single-center study showed that the use of SF was a promising supplement in the neurosurgical resection of BMs from lung cancer over standard resection techniques by increasing the EOR, reducing local recurrence, and improving patients’ prognosis. However, randomized prospective studies are needed to further assess the role of SF on the EOR and OS.

## Figures and Tables

**Figure 1 cancers-15-00882-f001:**
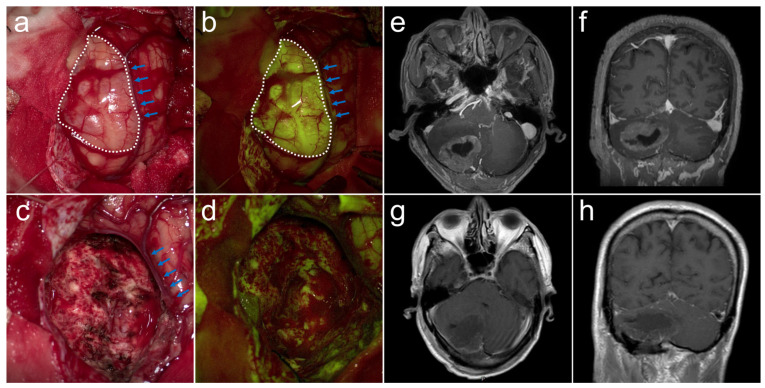
A patient in their 50s underwent SF-guided surgery for adenocarcinoma of the right cerebellar hemisphere. Microsurgical photographs showed the tumor (white dashed frame) located in the cerebellar hemispherical cortex and the draining vein (blue arrowheads) in (**a**) white light mode and (**b**) yellow light mode. The residual cavity after tumor resection in (**c**) white light mode and (**d**) yellow light mode and preservation of the draining vein. (**e**–**h**) Postoperative axial and coronal magnetic resonance imaging within 72 h showed complete removal of the tumor.

**Figure 2 cancers-15-00882-f002:**
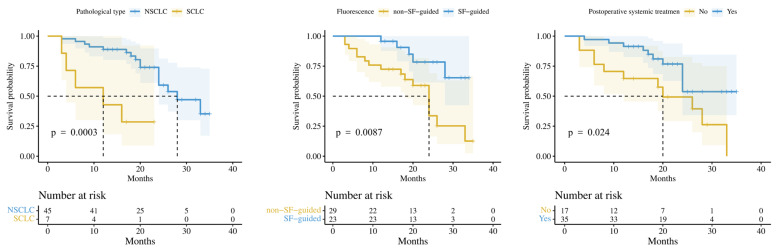
Kaplan–Meier survival curves in terms of pathological type (NSCLC vs. SCLC), fluorescence-guided surgery (SF-guided vs. non-SF-guided), and postoperative systemic treatment (Yes vs. No).

**Table 1 cancers-15-00882-t001:** Clinical baseline characteristics of patients with BMs from lung cancer.

Characteristics	SF-Guided *n* = 23 (%)	Non-SF-Guided *n* = 29 (%)	χ^2^	*p*-Value
Sex			0.015	0.904
Male	17 (73.9)	21 (72.4)		
Female	6 (26.1)	8 (27.6)		
Age at surgery			0.371	0.542
<60	14 (60.9)	20 (69.0)		
≥60	9 (39.1)	9 (31.0)		
Median	58	54		
Range	35–77	26–74		
Preoperative KPS			0.006	0.937
≥70	20 (87.0)	25 (86.2)		
<70	3 (13.0)	4 (13.8)		
Tumor location			0.090	0.764
Supratentorial	19 (82.6)	23 (79.3)		
Infratentorial	4 (17.4)	6 (20.7)		
Number of intracranial metastases			0.587	0.255
1	18 (78.3)	20 (69.0)		
2	3 (13.0)	5 (17.2)		
≥3	2 (8.7)	4 (13.8)		
Extracranial metastasis at surgery			1.123	0.289
Yes	7 (30.4)	13 (44.8)		
No	16 (69.6)	16 (55.2)		
Pathological type			0.006	0.937
NSCLC	20 (87.0)	25 (86.2)		
SCLC	3 (13.0)	4 (13.8)		
Postoperative systemic treatment (chemo-/targeted/immunotherapy)			0.777	0.378
Yes	14 (60.9)	21 (72.4)		
No	9 (39.1)	8 (27.6)		
Postoperative brain radiotherapy			0.055	0.815
Yes	8 (34.8)	11 (37.9)		
No	15 (65.2)	18 (62.1)		

SF, sodium fluorescein; KPS, Karnofsky Performance Status; NSCLC, non-small cell lung cancer; SCLC, small cell lung cancer.

**Table 2 cancers-15-00882-t002:** Comparison of extent of resection (EOR) and postoperative local recurrence.

Variables	SF-Guided *n* = 23 (%)	Non-SF-Guided *n* = 29 (%)	χ^2^	*p*-Value
EOR (RANO-BM)			4.038	0.044
N	20 (87.0)	18 (62.1)		
U and M	3 (23.0)	11 (37.9)		
Postoperative local recurrence			4.805	0.028
No	21 (91.3)	19 (65.5)		
Yes	2 (8.7)	10 (34.5)		

SF, sodium fluorescein; EOR, extent of resection; RANO-BM, Response Assessment in Neuro-Oncology Brain Metastases; N, no residual (no contrast enhancement on postoperative MRI); U, unmeasurable residual (<10 mm contrast-enhancing tumor remnant on postoperative MRI); M, measurable residual (>10 mm contrast-enhancing tumor remnant on postoperative MRI).

**Table 3 cancers-15-00882-t003:** Univariant and multivariate analysis for survival.

Independent Variables	Univariant Analysis		Multivariate Analysis	
	Hazard Ratio (95%CI)	*p*-Value	Hazard Ratio (95%CI)	*p*-Value
Sex				
Male	1.00		1.00	
Female	0.26 (0.10–0.67)	0.047	0.42 (0.09–1.95)	0.27
Age				
<60	1.00			
≥60	2.25 (0.87–5.79)	0.050		
Preoperative KPS				
≥70	1.00			
<70	1.00 (0.23–4.33)	0.994		
Tumor location				
Supratentorial	1.00			
Infratentorial	0.62 (0.18–2.09)	0.503		
Number of intracranial metastases				
1	1.00			
2	1.19 (0.37–3.84)	0.761		
≥3	1.57 (0.43–5.65)	0.418		
Extracranial metastasis at surgery				
Yes	1.00			
No	0.94 (0.39–2.24)	0.885		
Fluorescence				
SF-guided	1.00		1.00	
Non-SF-guided	3.43 (1.46–8.07)	0.009	3.92 (1.34–11.47)	0.013
EOR (RANO-BM)				
N	1.00			
U and M	2.17 (0.72–6.49)	0.074		
Pathological type				
NSCLC	1.00		1.00	
SCLC	5.05 (0.81–31.36)	0.000	5.18 (1.58–16.99)	0.007
Postoperative systemic treatment (chemo-/targeted/immunotherapy)				
Yes	1.00		1.00	
No	2.54 (1.00–6.44)	0.024	2.90 (1.13–7.49)	0.027
Postoperative brain radiotherapy				
Yes	1.00			
No	1.84 (0.76–4.48)	0.216		
Postoperative local recurrence				
No	1.00		1.00	
Yes	2.64 (0.99–7.02)	0.018	1.33 (0.51–3.44)	0.557

KPS, Karnofsky Performance Status; SF, sodium fluorescein; EOR, extent of resection; RANO-BM, Response Assessment in Neuro-Oncology Brain Metastases; N, no residual (no contrast enhancement on postoperative MRI); U, unmeasurable residual (<10 mm contrast-enhancing tumor remnant on postoperative MRI); M, measurable residual (>10 mm contrast-enhancing tumor remnant on postoperative MRI); NSCLC, non-small cell lung cancer; SCLC, small cell lung cancer.

## Data Availability

The raw and/or analyzed datas of the study are available from the corresponding author.

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
