# Peer review of "Sodium Fluorescein-Guided Surgery for Resection of Brain Metastases from Lung Cancer: A Consecutive Case Series Study and Literature Review"

_cancers, 2023, doi:10.3390/cancers15030882_

Round 1

Reviewer 1 Report

In this retrospective study by Cheng and colleagues, the application of sodium fluorescein in resection of mostly solitary metastatic lung cancer to the brain was examined compared to a similar cohort without fluorescein. Their protocol included administering fluorescein intravenously after induction. Interestingly, neuronavigation was only used for planning the craniotomy and localizing the tumor after dural opening, but not to guide resection in the fluorescein free cohort. Their primary endpoint was extent of resection, using RANO-BM definitions, although interestingly they combined measurable residual with unmeasurable residual and left no residual as a standalone group. They used conventional univariate and multivariate analysis to evaluate demographic group differences as well as to look at survival. In this small study of 23 fluorescein guided and 29 non fluorescein guided resections, the groups were mostly similar, although the non-fluorescein group has numerically more intracranial disease burden and higher systemic disease at surgery, although this was not statistically significant. They found that the fluorescein guided group had a higher proportion of no residual disease to measurable or unmeasurable disease compared to the non-fluorescein group and a lower rate of local recurrence.  Their multivariate analysis suggests a modest survival benefit with fluorescein, although a small cell diagnosis and lack of postoperative adjuvant therapies carried the greatest hazard for mortality. They claim in the discussion that the fluorescein group’s improved survival was due to higher extent of resection and local recurrence rate, although local recurrence was not significant in the multivariate analysis, calling this explanation into question.

Many metastatic brain tumors typically have a well-defined border, making the value of fluorescein inconstant.  However, arguably, its low cost makes it an acceptable adjunct. The authors’ statement that they did not use neuronavigation to guide the tumor resection other than planning the craniotomy and corticotomy is interesting, as I do wonder if the extent of resection in the non-fluorescein group would be higher if navigation was used to confirm adequately reaching the margins of the tumor and therefore lower the extent of resection benefit seen in the fluorescein group. I also question the chi-square analysis grouping measurable and unmeasurable disease and leaving no residual as a stand-alone; if no residual and unmeasurable were grouped together with the main test variable being residual disease, I again think the effect of fluorescein would statistically be reduced. This study is likely underpowered to legitimately say fluorescein has a survival benefit and some of these conclusions may represent Type 1 statistical error; the fact that local recurrence was not significant on the multivariate analysis stands athwart the claims of survival benefit for fluorescein. Moreover, the non-fluorescein group had higher degree of systemic disease burden and greater number of intracranial lesions compared to the fluorescein group that may in part explain the worse survival. Some of these issues must be addressed in a limitations section. Additionally, there are English grammar errors 

Author Response

Dear reviewer,

Thanks for your pertinent advice.

We would like to add and clarify the following:

  1. In fact, our expression in the manuscript was "Neuronavigation was used only for craniotomy planning, and cerebrocortical tumor localization, but not for resection control. ", referring to the SF-guided group. This was done to investigate the actual role and effectiveness of SF in guiding tumor resection and to exclude the influence of other intraoperative imaging tools. In the non-SF-guided group, navigation or intraoperative ultrasound was used to determine and guide tumor resection in most cases.
  2. As you said, if the grouping method of no residual+unmeasurable residual vs measurable residual is used, there is no statistical difference in EOR between the SF group and the non-SF group (p=0.42). We considered that no residual might be more convincing than no residual+unmeasurable residual to represent the maximum extent of resection.
  3. The chi-square test showed that the p-values of the SF-guided group and the non-SF-guided group in the degree of tumor resection were very close to 0.05 (0.044). The fact that we did not strictly use neuronavigation or intraoperative ultrasound to test excision extent in all non-SF-guided group cases may have contributed to the p < 0.05. As stated, this is a limited retrospective study with a limited level of evidence to conclude, and we are planning to conduct a rigorously designed randomized controlled trial to further clarify this issue.
  4. As you noted, cases in the non-SF-guided group had a higher number of intracranial and extracranial metastases, which may have resulted in poorer survival in this group. We will further explain this phenomenon and explore the role of SF in improving patient survival in the discussion section. We also emphasize at the end of the paper that more rigorously designed prospective clinical trials are needed to clarify this still controversial topic.

We appreciate again for your important review work and hope you will find our reply satisfactory.

Best wishes!

Haifeng Yang,M.D. 
yanghaifeng@cqu.edu.cn
Department of Neuro-Oncology, Chongqing University Cancer Hospital, Chongqing, China.
No.181, Hanyu Road, Shapingba District, Chongqing, China.
Fax: +86 02365075681

Reviewer 2 Report

The authors presented a retrospective analysis of the use of sodium fluorescein (SF) in brain metastases from lung cancers. A total of 52 patients were enrolled, of which 23 received SF-guided surgery and 29 did not. The authors evaluated that the use of SF positively affects the EOR, and reduces local recurrence. The value of the present series is its case-control structure, and the homogeneity of patients included. Therefore, the manuscript could represent a valuable addition to the current literature. I suggest adding a very large study on the use of SF in brain tumors (Falco J et al, Fluorescein Application in Cranial and Spinal Tumors Enhancing at Preoperative MRI and Operated With a Dedicated Filter on the Surgical Microscope: Preliminary Results in 279 Patients Enrolled in the FLUOCERTUM Prospective Study. Front Surg. 2019 Aug 13;6:49. doi: 10.3389/fsurg.2019.00049) considering that it confirms the usefulness of SF in brain metastases.

Could the authors better specify how was the dichotomization into the two groups performed? In other words, only the preference of the surgeon was the main determinant of the choice?

Author Response

Dear reviewer,

Thanks for your pertinent advice.

1. We've already added the article as one of the references.

2. Since sodium fluorescein was introduced in our department at the end of 2019 and was not routinely used in BMs surgery at that time, the selection of cases was fairly random. As a new drug and technology (for our department), the selection of cases mainly depended on the patient's will (signing informed consent) and the surgeon's judgment (location, size, whether it is located in the functional areas, whether it is close to important blood vessels, etc.).
This study is a retrospective study and its conclusions are limited. Fortunately, based on the results of this study, we are now planning a rigorously designed prospective study to more definitively answer questions about the use of sodium fluorescein in BMs.

We appreciate again for your important review work.

Best wishes!

Haifeng Yang,M.D. 
yanghaifeng@cqu.edu.cn
Department of Neuro-Oncology, Chongqing University Cancer Hospital, Chongqing, China.
No.181, Hanyu Road, Shapingba District, Chongqing, China.
Fax: +86 02365075681